# Associations between Motor Competence, Physical Activity and Sedentary Behaviour among Early School-Aged Children in the SELMA Cohort Study

**DOI:** 10.3390/children11060616

**Published:** 2024-05-21

**Authors:** Johanna Delvert, Heléne V. Wadensjö, Carl-Gustaf Bornehag, Sverre Wikström

**Affiliations:** 1Department of Health Sciences, Karlstad University, 651 88 Karlstad, Sweden; helene@kau.se (H.V.W.); carl-gustaf.bornehag@kau.se (C.-G.B.); sverre.wikstrom@regionvarmland.se (S.W.); 2Center for Clinical Research, Region Värmland County Council, 651 82 Karlstad, Sweden; 3Department of Environmental Medicine and Public Health, Icahn School of Medicine at Mount Sinai, New York, NY 10029, USA; 4School of Medical Sciences, Örebro University, 701 82 Örebro, Sweden

**Keywords:** exercise, fundamental motor skills, health promotion, sex differences

## Abstract

Low motor competence (MC) has been associated with lower physical activity (PA) and long-term health risks in children. Less is known about sex-specific patterns and associations during early school age. The aim of this study was to explore how motor difficulties are associated with PA levels, screen time, and organised sports participation (OSP). Data from 479 children, seven years of age, participating in the Swedish Environmental, Longitudinal, Mother and child, Asthma, and allergy (SELMA) pregnancy cohort study were used. MC and activity-related outcomes were assessed with questionnaires answered by parents. Associations between MC and outcomes were evaluated using logistic regression models adjusted for sex, overweight, and parental education level. Sex differences were investigated with interaction analyses and in stratified models. Children with motor difficulties had the same level of PA as their peers, but more screen time and lower OSP. Compared with children with normal MC, boys with motor difficulties had lower rates of OSP, but girls did not. This indicates that the identification and compensatory support for motor difficulties for boys at an early age, as well as the development of inclusive leisure time activities, are of importance to facilitate health-promoting activities on equal terms.

## 1. Introduction

The health benefits of increased physical activity (PA) and reduced sedentary behaviour are well documented and include improved physical fitness, better cardio-metabolic and bone health, reduced adiposity, and improvement in cognitive outcomes [1]. To establish healthy activity-related behaviours in childhood, it is essential to introduce regular PA, active transporting, sports participation, and play, and to reduce sedentary behaviour and screen time [1]. Swedish and other European data indicate that less than 50% of the young population reaches the recommended levels of daily PA [2,3,4]. Boys are more active than girls and activity levels for children decline at around six to seven years of age [2]. From a public health perspective, system changes are warranted to change this trend. Organised sports participation (OSP) is associated with higher overall levels of PA in both pre-school and school-aged children [5,6]. In Sweden, 75% of 11–15-year-old children participate in organised sports. After the age of 15, participation in organised sports declines [7,8]. The early development of motor competence (MC) is important not only for increased levels of PA and cardiorespiratory fitness, but also for many aspects of mental and social development. Positive experiences of inclusion are important in order to continue with an active lifestyle [9,10]. Children with MC below the 5th percentile often meet the criterion for developmental coordination disorder (DCD), which is a defined neurodevelopmental disorder affecting motor coordination and the development of motor skills [11]. However, less severe motor difficulties, i.e., MC below the 15th percentile, may impede participation in PA and recreation, and should be considered for support [12]. There is some evidence for a positive relationship between higher MC levels and OSP [13]. Lower MC levels, on the other hand, increase the risk of disappointments in relation to sports and play [9,14]. Both increased PA levels and a diversity of offered sports and play activities seem to facilitate the development of MC [15]. In order to build support for children with motor difficulties, more research is needed on the relation with health-promoting activity-related behaviours. Previous research indicates that the age where activity levels start to decline, as well as sex differences, needs to be further investigated if the consequences of low MC should be prevented. Therefore, the aim of the present study was to investigate how motor difficulties are associated with physical activity, screen time, and organised sports participation at seven years of age.

## 2. Materials and Methods

The current cross sectional analyses were based on data from the ongoing birth cohort Swedish Environmental, Longitudinal, Mother and child, Asthma, and allergy (SELMA) study (N = 1954) described by Bornehag et al. [16]. The SELMA children were invited to a follow up at seven years of age, where N = 1006 participated. The data for this current study were collected from this health and development examination of the children and questionnaires answered by the accompanying parent. MC was, in this study, defined as the degree of proficient performance in a broad range of motor skills used in everyday activities [17], and was evaluated using the instrument Five-To-Fifteen-Revised (5-15R), a parental questionnaire for the evaluation of development and behaviour in children [18]. The instrument consists of statements about the child that can be answered with “Does not apply” (0 points), “Applies sometimes or to some extent” (1 point), or “Definitely applies” (2 points). The calculated mean score from the domains of fine and gross motor skills were used. Children with motor difficulties were defined in accordance with the instrument with sex-specific cut offs, with those with a mean value above the 90th percentile belonging to a risk group for possible DCD [12,18,19].

Parents’ estimations of the children’s activity levels were collected from a health questionnaire (Figure 1). The three main outcomes used in this study were estimated daily PA (h/day), estimated screen time (h/day), and child’s participation in organised sport held by a leader (yes/no). The parents were asked about their children’s participation in physical education at school (No; Yes, but seldom; Yes, once a week; Yes, several times a week).

Height and weight were measured as previously described [20] and the children were categorised as normal weight or overweight/obese based on age- and sex-specific ISO-BMI [21]. Information on parental education level (for adjustment of analyses) was collected from the self-administrated questionnaires completed during early pregnancy for both mothers and fathers. Education level was categorised in two groups with high school or below categorised as a low education level, and college or university as a high education level.

Study group characteristics with group comparisons are shown as mean values and percentages. Differences between the study sample and the remaining cohort were analysed for the study variables, utilising the Mann–Whitney U test. Differences in PA levels and screen time between children with motor difficulties vs. higher motor competence were also analysed with the Mann–Whitney U test. The Chi-squared test was used for the corresponding comparison of OSP. Due to skewed distributions of data, the activity-related outcomes were dichotomised for use in the logistic regression models, investigating motor difficulties as a predictor, and also adjusting for sex, overweight/obesity, and parental education level. PA and screen time were then first grouped into tertiles, where the lowest and highest tertiles were compared in order to establish relevant contrasts. In addition, interaction between motor difficulties and sex was assessed in the models to investigate whether motor difficulties affect boys and girls differently. A significant interaction term (*p* ≤ 0.05) was considered evidence of a sex difference. As a result of the interaction analyses, the regression analyses were post hoc stratified by sex.

Data were analysed using IBM SPSS Statistics for Windows 27.0, and *p*-values ≤ 0.05 were considered significant.

## 3. Results

In total, 1006 children participated in the follow up. Of these, 961 completed the examinations, 852 had complete data for PA variables, 502 had complete data from the 5-15R motor competence questionnaire, and 479 had complete data for all study variables (including adjustment variables).

Comparisons between the study sample and children excluded due to incomplete data showed no differences in mean PA (3.37 vs. 3.64 h/day), mean screen time (2.62 vs. 2.69 h/day), OSP (82.3 vs. 81.2%), overweight/obesity (18.4 vs. 18.7%), high parental education (72.7 vs. 65.8%), or male sex (51.6 vs. 50.8%) (*p* = non-significant for all comparisons).

Table 1 shows descriptive characteristics with group comparisons. Boys had higher PA levels (3.56 vs. 3.16 h/day) and more screen time (2.79 vs. 2.43 h/day) compared to girls (*p* < 0.05), but boys’ and girls’ OSP was equal (>80%). All children participated in physical education at school.

Overweight/obesity was not overall associated with PA, screen time, or OSP (Table 1); neither was associated with MC, and were present in 102/480 (21%) of children with normal MC and in 15/52 (29%) of children with motor difficulties (*p* = 0.22 for difference in proportions). Nearly two thirds of the children had at least one parent with a high education level, which had significant associations with all outcome variables (*p* < 0.01) (Table 1).

### Motor Difficulties, Physical Activity, and Sedentary Behaviour

Among the boys, 32/278 (12%) met the sex-specific criteria for motor difficulties, and in girls, the corresponding numbers were 20/264 (8%), *p* = 0.15 for the comparison. There were no differences in PA levels among children with motor difficulties as compared with children with higher MC, but screen time was increased and OSP was lower (Table 1).

Crude and adjusted logistic regression models (Table 2), comparing children with high versus low overall PA levels, showed no associations with MC. However, children with motor difficulties had more screen time, with a borderline significant association (Table 2). MC was a significant predictor of OSP, where difficulties were associated with lower participation even after adjusting for covariates (Table 2). The analyses of interaction effects between motor difficulties and sex, in relation to the three outcome measures, respectively, showed no interaction for PA or screen time. For OSP, there was as a significant interaction term (*p* = 0.04), indicating a sex difference and showing that boys with difficulties were less likely to participate in organised sports. Consequently, the models were stratified by sex.

Stratified regression analyses (Table 3) showed that boys with motor difficulties, compared to boys with normal MC, were about 22% less likely to engage in sports when adjusted for overweight/obesity and parental education level. Such an association was not seen among girls. The stratified analyses identified no significant association between MC and PA or screen time.

## 4. Discussion

The current study investigated the associations between motor difficulties and physical activity levels, screen time, and organised sports participation in early school-aged children. Although children with and without motor difficulties had the same levels of overall PA, motor difficulties were associated with increased screen time. Furthermore, this study showed that both boys and girls participated in equally high levels of organised sports. However, boys with motor difficulties were less engaged in organised sports compared to their peers.

At the same time, boys were both more active and had more sedentary time compared to girls, which is in line with previous research [2,7]. Previous research shows that motor difficulties normally precede inactivity and that the negative health consequences of poor MC develop over time [12,22,23,24]. The present results illustrate no association between motor difficulties and overall PA by the age of seven years, which is compatible with previous findings . For example, Lopes et al. [25] found no associations between motor coordination, physical fitness, and physical activity among six-year-old children. However, their longitudinal study showed that high MC attenuated the common age-related decrease in PA [25]. We find this interesting, as the interaction analysis in our present results indicated the exclusion of boys with motor difficulties from organised sports, despite the high overall PA levels and OSP rates in the overall study sample. Our findings might contribute important insights when targeting the age-related decline in health-promoting PA among children. The findings from early age are inconsistent, as Wälti et al. found that both low gross MC and overweight had a negative impact on PA and OSP in six- to eight-year-old children across Europe [13]. This inconsistency might be due to the different measures of MC [9]. It could also be explained from a bio-ecological perspective, where interactions between personal characteristics and the environment cater to children’s abilities and needs differently [9,26]. The present study contributes to the evidence base with knowledge from the unique Swedish context, where there are higher rates of OSP.

Sedentary behaviour is pinpointed as a risk factor that negatively affects the development of MC [27]. In our present study, screen time was assessed as a measure of sedentary behaviour and we found associations with sex, MC, and parental education level. Children with motor difficulties had 30 more minutes of screen time every day, even though the associations from the regression models were only borderline significant. Our findings are in line with previous research showing that lower MC is associated with more sedentary and screen time [27]. Since screen time may contribute to a lack of experiences of activity that promote MC development [27], we suggest that it should be considered in preventive work against the consequences of motor difficulties.

There is evidence that high body weight and low MC have a negative bidirectional association over time [24]. In our study, there were no associations between overweight/obesity and activity outcomes, nor between overweight/obesity and MC. As Stodden et al. [10] outlined in a conceptual model, negative experiences due to low MC will increase the risk of becoming inactive and developing obesity. The authors pinpoint the transition from early to middle school as a vulnerable time, as the children more adequately compare their motor skills with peers, and shortcomings might lead to disengagement. Early interventions intending to improve MC have shown good results [28]. These findings, combined with the present results, indicate that health-promoting activities should be introduced before such negative experiences occur. This can also contribute to the prevention of future non-communicable diseases associated with both inactivity and obesity.

Since OSP facilitates health-promoting and diverse PA over time [5,15], equal access is important. Unlike most previous studies [6], we found equal OSP for girls and boys. This might be due to the regional (Swedish) context, where national data have shown that 79% of the boys and 82% of the girls aged 11 to 12 years participate in leisure time sport activities [5]. Sweden has higher OSP rates, with smaller sex differences, compared to many other European countries [6]. Despite the overall equal OSP for girls and boys, the association between motor difficulties and OSP differed between the sexes in our study. Boys with motor difficulties were 22% less likely to be engaged in OSP than boys with normal MC in the adjusted models. Girls’ OSP was not associated with MC. Our study cannot explain why boys, but not girls, with motor difficulties had lower OSP. Previous studies of children about the same age indicate that boys’ higher competence in object skills promote inclusion in ball sports. On the other hand, girls’ higher competence in self-movement (i.e., body control) either promoted individual sports participation or increased exclusion from sports participation. This is of possible importance for the identified differences in OSP [9,13]. Together with our findings, this implies that more research is needed on sex differences in relation to MC and OSP. For the prevention of unequal access to health-promoting leisure time activities, future studies may consider resilience factors and lack of support, respectively, as they may also differ between boys and girls [9,13].

Our study contributes data from a Swedish cohort, assessed during early school age, when patterns of activity differ between countries and PA levels start to decline [2,13]. The present results indicate that the community offers inclusive activities, since 82% of both boys and girls are engaged in organised sports. However, both SES (i.e., parental education) and motor difficulties were associated with 15% lower rates of participation in organised sports, pinpointing that equal access is still not ensured. As discussed, the support for an association between low MC and lower OSP is growing, but there are also divergent findings [29], including subgroups of children who do not follow the expected patterns as they participate in sports despite low perceived and measured MC. This, in turn, supports the conclusion that OSP becomes a parental responsibility, especially for building resilience in children with difficulties [30].

We acknowledge several methodological limitations. The disadvantages associated with self-reported PA questionnaires are well documented, but for convenience, they are commonly used [31]. The choice to ask the parents instead of the children was made due to our judgement that it would have been even more difficult for the children at this young age to estimate time. Our question about OSP was, however, binary and had a high response rate (99% in the present sample), indicating that it was perceived as being easily answered. The answering rates on the screen time questions were also relatively high (96%), possibly indicating parental knowledge about their children’s habits at this age. The question about the daily amount of PA had the lowest answering rate (93%), and we acknowledge that parents may have found it difficult to interpret the question or to estimate the measure. Parents were asked to estimate their child’s daily time spent engaging in activities with an increased heart rate (MVPA) (Figure 1). Accelerometry-measured data for MVPA on children of the same age report an average of 45–60 min [2], in contrast to our PA estimates of 3–4 h/day. Even though measured data tend to underestimate time spent on an activity [31], our estimates are still high, leading us to suspect that the parents included lower-intensity activities as well. From a public health perspective, time spent engaging in muscle-strengthening, play, and low-intensity activities may also be of great importance for the development of MC [1,31], therefore constituting a relevant outcome measure; however, our results on PA must be interpreted with great care.

Due to the verified interaction by sex in the relation between motor difficulties and OSP (i.e., a statistically significant sex difference), we found it important to report sex-stratified analyses, but the smaller sample sizes must then be considered, which is also the case of multivariate analyses comparing high vs. low activity and screen time groups. This dichotomisation may have masked variations within the dataset and might be a reason for the borderline association between MC and screen time in the adjusted models.

The 5-15R aims to detect neurodevelopmental impairments such as ADHD and DCD. It has acceptable psychometric properties and good validity and reliability [19,32,33,34], and we consider the use of it, with a cutoff at the 90th percentile, as an appropriate screening method for motor difficulties [12]; however, it cannot replace a full clinical diagnostic evaluation. Parental evaluations, though, may not capture the full spectrum of motor challenges as experienced by children. Exclusion due to incomplete 5-15R questionnaire data explains the considerable loss of participants in this study. This was the consequence of a missing page in the distributed questionnaires, and we acknowledge the effect on the statistical power of the study. However, the mistake occurred at random. Therefore, we find that any systematic bias and selection bias is very unlikely, which is further supported by the absence of differences between our sample and the remaining cohort regarding other study variables.

Finally, adjusting the present analyses for sex, overweight/obesity, and parental education level is supported by previous literature, but there might still be unmeasured confounders that influence the observed associations. The parents from the SELMA cohort had a higher education level than the general population [16]. This may impact external validity, but we find no reason to question the findings of a sex-specific association between MC and OSP. Rather, it may be suspected that the high SES in the present sample contributed to overall high rates of OSP, possibly also compensating for exclusion effects due to motor difficulties, therefore constituting a risk of underestimated associations between motor difficulties and OSP.

Due to the cross sectional design of the present study, we cannot draw any conclusions on causality behind the correlation between motor difficulties, OSP, and health, and cannot explain why the participation differed.

In order to enable the prevention of unequal access to health-promoting leisure time activities, we suggest that future studies should further investigate the inequalities and lack of support that explain why children with low SES or boys with low MC may also have lower OSP. Children’s own thoughts and experiences may also be crucial for the development of inclusive environments.

## 5. Conclusions

Motor difficulties had a negative impact on organised sports participation, adjusted for overweight and parental education level. Interestingly enough, this was only found among boys. There was no association between motor competence and overall physical activity, and only a weak association with screen time. Identification and compensatory support for motor difficulties during early age, as well as the development of inclusive leisure time activities, may, therefore, be important to facilitate health-promoting physical activity on equal terms.

## Figures and Tables

**Figure 1 children-11-00616-f001:**
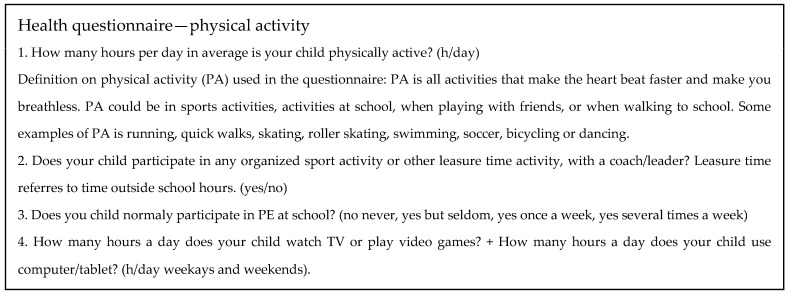
Health questionnaire (answered by parents) at seven years of age.

**Table 1 children-11-00616-t001:** Study group characteristics with group comparisons.

	N (%)	Physical Activity h/Day (Mean ± SD)	(*p*)	Screen Time h/day (Mean ± SD)	(*p*)	Organised Sports Participation N (%)	(*p*)
Total sample	479	3.37 ± 2.14 Range: 0–15 h		2.62 ± 1.20 Range: 0.14–8.57 h		394 (82.3)	
Sex: Girls	232 (48.4)	3.16 ± 2.08	0.005	2.43 ± 1.06	0.001	191 (82.3) 203 (82.2)	0.97
Boys	247 (51.6)	3.56 ± 2.18		2.79 ± 1.29			
ISO-BMI: normal overweight/obesity	391 (81.6) 88 (18.4)	3.42 ± 2.18 3.15 ± 1.93	0.28	2.57 ± 1.18 2.84 ± 1.24	0.06	324 (82.9) 70 (79.5)	0.46
*5–15 questionnaire*							
Motor skills: normal difficulties	436 (91) 43 (9)	3.34 ± 2.08 3.68 ± 2.69	0.67	2.57 ± 1.18 3.08 ± 1.33	0.02	364 (83.5) 29 (69)	0.01
*Parental variables*							
Education level:			<0.001		<0.001		<0.001
low	131 (27.3)	3.90 ± 2.38		2.94 ± 1.17		93 (71.0)	
high	348 (72.7)	3.17 ± 2.01		2.50 ± 1.19		301 (86.5)	
		Mann–Whitney U		Mann–Whitney U		Chi^2^	

**Table 2 children-11-00616-t002:** Associations between motor difficulties and activity-related outcomes.

	Physical Activity (Low vs. High)	*p*	Screen Time (Low vs. High)	*p*	Organised Sports (Yes/No)	*p*
Crude OR (CI)	1.30 (0.61–2.76)	0.49	2.12 (0.97–4.64)	0.06	0.44 (0.22–0.89)	0.02
Adjusted OR (CI)	1.25 (0.57–2.73)	0.58	2.15 (0.95–4.87)	0.07	0.40 (0.19–0.83)	0.01

N = 479. Binary logistic regression—low versus high groups. Adjusted for sex, overweight/obesity, and parental education.

**Table 3 children-11-00616-t003:** Associations between motor difficulties and activity-related outcomes, stratified by sex.

		Physical Activity (Low vs. High)	*p*	Screen Time (Low vs. High)	*p*	Organised Sports (Yes/No)	*p*
Crude OR (CI)	Girls	1.84 (0.54–6.33)	0.33	2.28 (0.66–7.84)	0.19	1.54 (0.34–7.06)	0.58
	Boys	0.96 (0.37–2.50)	0.93	1.85 (0.66–5.18)	0.24	0.24 (0.10–0.57)	0.001
Adjusted OR (CI)	Girls	1.94 (0.55–6.78)	0.30	2.61 (0.73–9.33)	0.14	1.40 (0.230–6.58)	0.67
	Boys	0.94 (0.35–2.55)	0.91	1.89 (0.65–5.47)	0.24	0.22 (0.09–0.54)	<0.001

N (girls) = 232. N (boys) = 247. Binary logistic regression—low versus high groups. Adjusted for overweight/obesity and parental education.

## Data Availability

Basal data on PA levels, screen time, and OSP can be made available to researchers upon request (subject to a review of secrecy). Requests for data should be directed to the Head of Department of Health Sciences, Karlstad University. However, according to the Ethical Review Board decision and obtained personal consent, data on motor competence, constituting clinical data, cannot be made freely available as they are subject to secrecy in accordance with the Swedish Public Access to Information and Secrecy Act [OSL 2009:400]. Unique combinations of such 5-15R data may make a study participant identifiable; consequently, no such data will be shared.

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
