# Peer review of "Associations between Motor Competence, Physical Activity and Sedentary Behaviour among Early School-Aged Children in the SELMA Cohort Study"

_children, 2024, doi:10.3390/children11060616_

Round 1
Reviewer 1 Report
Comments and Suggestions for Authors
Associations Between Motor Competence, Physical Activity and Sedentary Behaviour among Early School-aged Children in the SELMA Cohort Study
The research is important because it reveals how motor competence correlates with physical activity, screen time, and organized sports in children. The findings emphasize the importance of early identification of motor difficulties and providing appropriate support, especially for boys. Additionally, it highlights the significance of inclusive leisure activities in ensuring equal opportunities for participation in physical activity. The study is important because it examines an age group (early school-age) that represents the phase of school integration after preschool. This period is characterized by the transition from free and unstructured activities to school learning and a sedentary lifestyle, posing new challenges for children, parents, and educators alike.
The strength of this scientific article's introduction lies in its comprehensive overview of the importance of physical activity (PA) and the challenges associated with sedentary behavior in childhood. It effectively highlights the health benefits of PA and the concerning trend of low activity levels among children, supported by relevant statistics. Furthermore, it discusses the association between organized sports participation (OSP), motor competence (MC), and overall activity levels, providing a clear rationale for the study's focus on investigating these relationships. Finally, it outlines the specific aim of the study, which is to examine how motor difficulties impact various aspects of physical activity and sports participation in seven-year-old children.
One potential criticism of this research methodology is the reliance on parental estimations and subjective assessments of children's motor competence and activity levels. The use of parental questionnaires, while common in research involving young children, may introduce biases or inaccuracies due to parental perception or interpretation of the child's abilities and behaviors. Additionally, the definition of motor difficulties based on specific cut-offs from the Five-To-Fifteen-Revised (5-15R) instrument may not capture the full spectrum of motor challenges experienced by children. Furthermore, the categorization of children into tertiles for physical activity and screen-time may oversimplify the data and potentially mask important variations within each group. Finally, while logistic regression analysis and adjustment for potential confounding variables such as sex, overweight/obesity, and parental education level are essential for controlling for relevant factors, there may still be unmeasured confounders that influence the observed associations. Overall, while the methodology provides valuable insights, these limitations should be considered when interpreting the results. True, while the study lists its limitations, it might also be beneficial to refer to the limitations related to what I have described.
The article examines a very important topic area. After a more detailed explanation of the methodological limitations, I professionally recommend its publication.
Author Response
See attatched file

Reviewer 2 Report
Comments and Suggestions for Authors
Dear Authors,
the research presented in the article is relevant in today's context of sedentary behaviour.
The abstract is submitted in compliance with the requirements.
The introduction of the article is concise, specific and clear. It presents both the research problem and the clear objective.
The methodology part is presented, but it would be good to know more about the selection of subjects. This part would also be strengthened by more specific data analysis tests.
Research results are presented clearly, informative tables are provided.
The strongest part of this article is the discussion, which analyzes the results in depth.
The conclusion is concrete, responding to the results of the research.
I suggest authors to manage the bibliography more carefully, taking into account the journal's requirements: articles are presented without a doi; missing links to online sources.
I think that the presented shortcomings are not essential, but their correction should contribute to the quality of the article
Author Response
See attatched file

Reviewer 3 Report
Comments and Suggestions for Authors
This is an interesting study on boys and girls with motor difficulties. I felt that it had a certain level of quality, but I would like you to reconsider the following points.
1. On line 96, physical activity and sedentary behavior are dichotomized into high and low, but there is no explanation for how they are dichotomized. Please add an appropriate explanation if necessary.
2. If you read lines 107 to 109, you will see that the number of subjects analyzed was reduced from the original 1,006 to a final 479. In particular, the number of subjects who could not be assessed for motor difficulties, which was the focus of this study, was drastically reduced to 502, roughly half the number. I think that the results and discussion should be done to see if there is any bias in the current study, such as the fact that many boys and girls with motor difficulties remain.
Author Response
See attatched file
